# Single Pass PCA of Matrix Products

**Shanshan Wu**
The University of Texas at Austin
shanshan@utexas.edu

**Srinadh Bhojanapalli**
Toyota Technological Institute at Chicago
srinadh@ttic.edu

**Sujay Sanghavi**
The University of Texas at Austin
sanghavi@mail.utexas.edu

**Alexandros G. Dimakis**
The University of Texas at Austin
dimakis@austin.utexas.edu

## Abstract

In this paper we present a new algorithm for computing a low rank approximation of the product $A^T B$ by taking only a single pass of the two matrices $A$ and $B$. The straightforward way to do this is to (a) first sketch $A$ and $B$ individually, and then (b) find the top components using PCA on the sketch. Our algorithm in contrast retains additional summary information about $A, B$ (e.g. row and column norms etc.) and uses this additional information to obtain an improved approximation from the sketches. Our main analytical result establishes a comparable spectral norm guarantee to existing two-pass methods; in addition we also provide results from an Apache Spark implementation[1] that shows better computational and statistical performance on real-world and synthetic evaluation datasets.

## 1 Introduction

Given two large matrices $A$ and $B$ we study the problem of finding a low rank approximation of their product $A^T B$, using only one pass over the matrix elements. This problem has many applications in machine learning and statistics. For example, if $A = B$, then this general problem reduces to Principal Component Analysis (PCA). Another example is a low rank approximation of a co-occurrence matrix from large logs, e.g., $A$ may be a user-by-query matrix and $B$ may be a user-by-ad matrix, so $A^T B$ computes the joint counts for each query-ad pair. The matrices $A$ and $B$ can also be two large bag-of-word matrices. For this case, each entry of $A^T B$ is the number of times a pair of words co-occurred together. As a fourth example, $A^T B$ can be a cross-covariance matrix between two sets of variables, e.g., $A$ and $B$ may be genotype and phenotype data collected on the same set of observations. A low rank approximation of the product matrix is useful for Canonical Correlation Analysis (CCA) [3]. For all these examples, $A^T B$ captures pairwise variable interactions and a low rank approximation is a way to efficiently represent the significant pairwise interactions in sub-quadratic space.

Let $A$ and $B$ be matrices of size $d \times n$ $(d \gg n)$ assumed too large to fit in main memory. To obtain a rank-$r$ approximation of $A^T B$, a naive way is to compute $A^T B$ first, and then perform truncated singular value decomposition (SVD) of $A^T B$. This algorithm needs $O(n^2 d)$ time and $O(n^2)$ memory to compute the product, followed by an SVD of the $n \times n$ matrix. An alternative option is to directly run power method on $A^T B$ without explicitly computing the product. Such an algorithm will need to access the data matrices $A$ and $B$ multiple times and the disk IO overhead for loading the matrices to memory multiple times will be the major performance bottleneck.

For this reason, a number of recent papers introduce randomized algorithms that require only a few passes over the data, approximately linear memory, and also provide spectral norm guarantees. The

key step in these algorithms is to compute a smaller representation of data. This can be achieved by two different methods: (1) dimensionality reduction, i.e., matrix sketching [15, 5, 14, 6]; (2) random sampling [7, 1]. The recent results of Cohen et al. [6] provide the strongest spectral norm guarantee of the former. They show that a sketch size of $O(\tilde{r}/\epsilon^2)$ suffices for the sketched matrices $\widetilde{A}^T\widetilde{B}$ to achieve a spectral error of $\epsilon$, where $\tilde{r}$ is the maximum stable rank of $A$ and $B$. Note that $\widetilde{A}^T\widetilde{B}$ is not the desired rank-$r$ approximation of $A^TB$. On the other hand, [1] is a recent sampling method with very good performance guarantees. The authors consider entrywise sampling based on column norms, followed by a matrix completion step to compute low rank approximations. There is also a lot of interesting work on streaming PCA, but none can be directly applied to the general case when $A$ is different from $B$ (see Figure 4(c)). Please refer to Appendix D for more discussions on related work.

Despite the significant volume of prior work, there is no method that computes a rank-$r$ approximation of $A^TB$ when the entries of $A$ and $B$ are streaming in a single pass [2]. Bhojanapalli et al. [1] consider a two-pass algorithm which computes column norms in the first pass and uses them to sample in a second pass over the matrix elements. In this paper, we combine ideas from the sketching and sampling literature to obtain the first algorithm that requires only a single pass over the data.

**Contributions:** We propose a one-pass algorithm SMP-PCA (which stands for Streaming Matrix Product PCA) that computes a rank-$r$ approximation of $A^TB$ in time $O((\text{nnz}(A) + \text{nnz}(B))\frac{\rho^2 r^3 \tilde{r}}{\eta^2} + \frac{nr^6\rho^4\tilde{r}^3}{\eta^4})$. Here $\text{nnz}(\cdot)$ is the number of non-zero entries, $\rho$ is the condition number, $\tilde{r}$ is the maximum stable rank, and $\eta$ measures the spectral norm error. Existing two-pass algorithms such as [1] typically have longer runtime than our algorithm (see Figure 3(a)). We also compare our algorithm with the simple idea that first sketches $A$ and $B$ separately and then performs SVD on the product of their sketches. We show that our algorithm *always* achieves better accuracy and can perform arbitrarily better if the column vectors of $A$ and $B$ come from a cone (see Figures 2, 4(b), 3(b)).

The central idea of our algorithm is a novel *rescaled JL embedding* that combines information from matrix sketches and vector norms. This allows us to get better estimates of dot products of high dimensional vectors compared to previous sketching approaches. We explain the benefit compared to a naive JL embedding in Figure 2 and the related discussion; we believe it may be of more general interest beyond low rank matrix approximations.

We prove that our algorithm recovers a low rank approximation of $A^TB$ up to an error that depends on $\|A^TB - (A^TB)_r\|$ and $\|A^TB\|$, decaying with increasing sketch size and number of samples (Theorem 3.1). The first term is a consequence of low rank approximation and vanishes if $A^TB$ is exactly rank-$r$. The second term results from matrix sketching and subsampling; the bounds have similar dependencies as in [6].

We implement SMP-PCA in Apache Spark and perform several distributed experiments on synthetic and real datasets. Our distributed implementation uses several design innovations described in Section 4 and Appendix C.5 and it is the only Spark implementation that we are aware of that can handle matrices that are large in both dimensions. Our experiments show that we improve by approximately a factor of $2\times$ in running time compared to the previous state of the art and scale gracefully as the cluster size increases. The source code is available at [18].

In addition to better performance, our algorithm offers another advantage: It is possible to compute low-rank approximations to $A^TB$ even when the entries of the two matrices arrive in some arbitrary order (as would be the case in streaming logs). We can therefore discover significant correlations even when the original datasets cannot be stored, for example due to storage or privacy limitations.

## 2 Problem setting and algorithms

Consider the following problem: given two matrices $A \in \mathbb{R}^{d \times n_1}$ and $B \in \mathbb{R}^{d \times n_2}$ that are stored in disk, find a rank-$r$ approximation of their product $A^TB$. In particular, we are interested in the setting where both $A$, $B$ and $A^TB$ are too large to fit into memory. This is common for modern large scale machine learning applications. For this setting, we develop a single-pass algorithm SMP-PCA that computes the rank-$r$ approximation without explicitly forming the entire matrix $A^TB$.

**Notations.** Throughout the paper, we use $A(i, j)$ or $A_{ij}$ to denote $(i, j)$ entry for any matrix $A$. Let $A_i$ and $A^j$ be the $i$-th column vector and $j$-th row vector. We use $\|A\|_F$ for Frobenius norm, and $\|A\|$ for spectral (or operator) norm. The optimal rank-$r$ approximation of matrix $A$ is $A_r$, which can be found by SVD. For any positive integer $n$, let $[n]$ denote the set $\{1, 2, \cdots, n\}$. Given a set $\Omega \subset [n_1] \times [n_2]$ and a matrix $A \in \mathbb{R}^{n_1 \times n_2}$, we define $P_\Omega(A) \in \mathbb{R}^{n_1 \times n_2}$ as the projection of $A$ on $\Omega$, i.e., $P_\Omega(A)(i, j) = A(i, j)$ if $(i, j) \in \Omega$ and 0 otherwise.

## 2.1 SMP-PCA

Our algorithm SMP-PCA (Streaming Matrix Product PCA) takes four parameters as input: the desired rank $r$, number of samples $m$, sketch size $k$, and the number of iterations $T$. Performance guarantee involving these parameters is provided in Theorem 3.1. As illustrated in Figure 1, our algorithm has three main steps: 1) compute sketches and side information in one pass over $A$ and $B$; 2) given partial information of $A$ and $B$, estimate *important* entries of $A^T B$; 3) compute low rank approximation given estimates of a few entries of $A^T B$. Now we explain each step in detail.

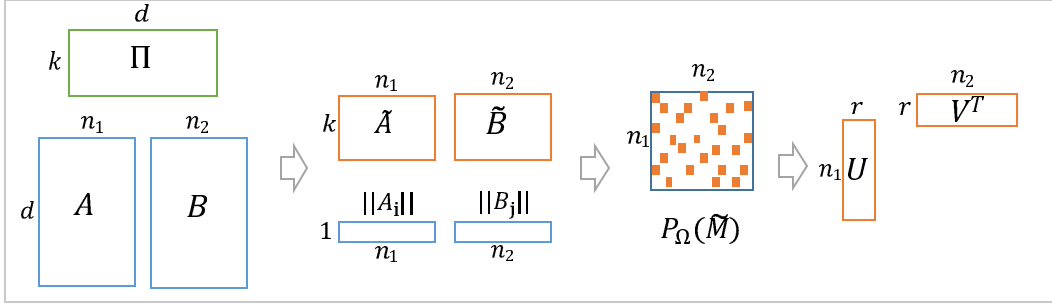

Figure 1: An overview of our algorithm. A single pass is performed over the data to produce the sketched matrices $\widetilde{A}$, $\widetilde{B}$ and the column norms $\|A_i\|$, $\|B_j\|$, for $i \in [n_1]$ and $j \in [n_2]$. We then compute the sampled matrix $P_\Omega(\widetilde{M})$ through a biased sampling process, where $P_\Omega(\widetilde{M}) = \widetilde{M}(i, j)$ if $(i, j) \in \Omega$ and zero otherwise. Here $\Omega$ represents the set of sampled entries. The $(i, j)$-th entry of $\widetilde{M}$ is given in Eq. (2). Performing matrix completion on $P_\Omega(\widetilde{M})$ gives the desired rank-$r$ approximation.

---

**Algorithm 1** SMP-PCA: Streaming Matrix Product PCA
___
1: **Input:** $A \in \mathbb{R}^{d \times n_1}$, $B \in \mathbb{R}^{d \times n_2}$, desired rank: $r$, sketch size: $k$, number of samples: $m$, number of iterations: $T$
2: Construct a random matrix $\Pi \in \mathbb{R}^{k \times d}$, where $\Pi(i, j) \sim \mathcal{N}(0, 1/k)$, $\forall (i, j) \in [k] \times [d]$. Perform a single pass over $A$ and $B$ to obtain: $\widetilde{A} = \Pi A$, $\widetilde{B} = \Pi B$, and $\|A_i\|$, $\|B_j\|$, for $i \in [n_1]$ and $j \in [n_2]$.
3: Sample each entry $(i, j) \in [n_1] \times [n_2]$ independently with probability $\hat{q}_{ij} = \min\{1, q_{ij}\}$, where $q_{ij}$ is defined in Eq.(1); maintain a set $\Omega \subset [n_1] \times [n_2]$ which stores all the sampled pairs $(i, j)$.
4: Define $\widetilde{M} \in \mathbb{R}^{n_1 \times n_2}$, where $\widetilde{M}(i, j)$ is given in Eq. (2). Calculate $P_\Omega(\widetilde{M}) \in \mathbb{R}^{n_1 \times n_2}$, where $P_\Omega(\widetilde{M}) = \widetilde{M}(i, j)$ if $(i, j) \in \Omega$ and zero otherwise.
5: Run WAltMin($P_\Omega(\widetilde{M}), \Omega, r, \hat{q}, T$), see Appendix A for more details.
6: **Output:** $\widehat{U} \in \mathbb{R}^{n_1 \times r}$ and $\widehat{V} \in \mathbb{R}^{n_2 \times r}$.
___

**Step 1: Compute sketches and side information in one pass over $A$ and $B$.** In this step we compute sketches $\widetilde{A} := \Pi A$ and $\widetilde{B} := \Pi B$, where $\Pi \in \mathbb{R}^{k \times d}$ is a random matrix with entries being i.i.d. $\mathcal{N}(0, 1/k)$ random variables. It is known that $\Pi$ satisfies an "oblivious Johnson-Lindenstrauss (JL) guarantee" [15][17] and it helps preserving the top row spaces of $A$ and $B$ [5]. Note that any sketching matrix $\Pi$ that is an oblivious subspace embedding can be considered here, e.g., sparse JL transform and randomized Hadamard transform (see [6] for more discussion).

Besides $\widetilde{A}$ and $\widetilde{B}$, we also compute the $L_2$ norms for all column vectors, i.e., $\|A_i\|$ and $\|B_j\|$, for $i \in [n_1]$ and $j \in [n_2]$. We use this additional information to design better estimates of $A^T B$ in the

next step, and also to determine *important* entries of $\widetilde{A}^T \widetilde{B}$ to sample. Note that this is the only step that needs one pass over data.

**Step 2: Estimate important entries of $A^T B$ by rescaled JL embedding.** In this step we use partial information obtained from the previous step to compute a few important entries of $\widetilde{A}^T \widetilde{B}$. We first determine what entries of $\widetilde{A}^T \widetilde{B}$ to sample, and then propose a novel rescaled JL embedding for estimating those entries.

We sample entry $(i, j)$ of $A^T B$ independently with probability $\hat{q}_{ij} = \min\{1, q_{ij}\}$, where

$$q_{ij} = m \cdot (\frac{\|A_i\|^2}{2n_2\|A\|_F^2} + \frac{\|B_j\|^2}{2n_1\|B\|_F^2}). \tag{1}$$

Let $\Omega \subset [n_1] \times [n_2]$ be the set of sampled entries $(i, j)$. Since $\mathbb{E}(\sum_{i,j} q_{ij}) = m$, the expected number of sampled entries is roughly $m$. The special form of $q_{ij}$ ensures that we can draw $m$ samples in $O(n_1 + m \log(n_2))$ time; we show how to do this in Appendix C.5.

Note that $q_{ij}$ intuitively captures important entries of $A^T B$ by giving higher weight to heavy rows and columns. We show in Section 3 that this sampling actually generates good approximation to the matrix $A^T B$.

The biased sampling distribution of Eq. (1) is first proposed by Bhojanapalli et al. [1]. However, their algorithm [1] needs a second pass to compute the sampled entries, while we propose a novel way of estimating dot products, using information obtained in the first step.

Define $\widetilde{M} \in \mathbb{R}^{n_1 \times n_2}$ as

$$\widetilde{M}(i, j) = \|A_i\| \cdot \|B_j\| \cdot \frac{\widetilde{A}_i^T \widetilde{B}_j}{\|\widetilde{A}_i\| \cdot \|\widetilde{B}_j\|}. \tag{2}$$

Note that we will not compute and store $\widetilde{M}$, instead, we only calculate $\widetilde{M}(i, j)$ for $(i, j) \in \Omega$. This matrix is denoted as $P_\Omega(\widetilde{M})$, where $P_\Omega(\widetilde{M})(i, j) = \widetilde{M}(i, j)$ if $(i, j) \in \Omega$ and 0 otherwise.

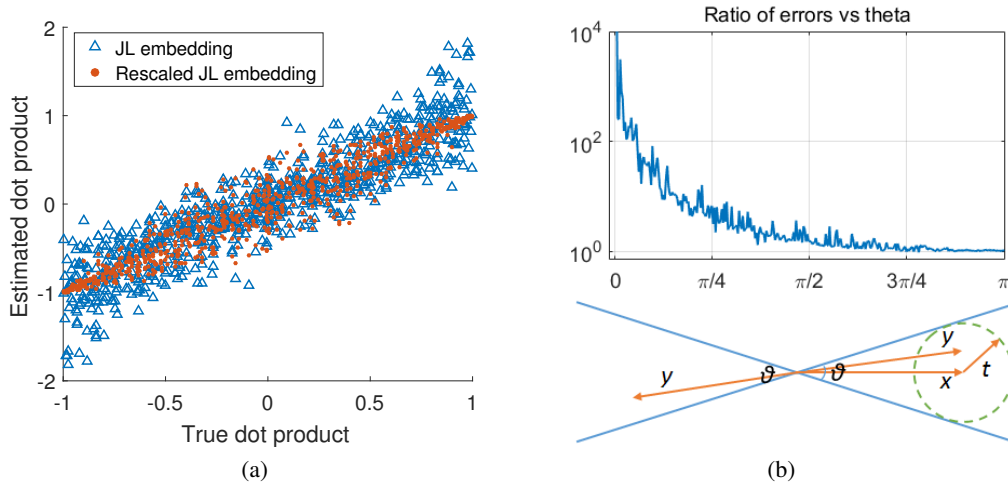

(a)                                        (b)

Figure 2: (a) Rescaled JL embedding (red dots) captures the dot products with smaller variance compared to JL embedding (blue triangles). Mean squared error: 0.053 versus 0.129. (b) Lower figure illustrates how to construct unit-norm vectors from a cone with angle $\theta$. Let $x$ be a fixed unit-norm vector, and let $t$ be a random Gaussian vector with expected norm $\tan(\theta/2)$, we set $y$ as either $x + t$ or $-(x + t)$ with probability half, and then normalize it. Upper figure plots the ratio of spectral norm errors $\|A^T B - \widetilde{A}^T \widetilde{B}\| / \|A^T B - \widetilde{M}\|$, when the column vectors of $A$ and $B$ are unit vectors drawn from a cone with angle $\theta$. Clearly, $\widetilde{M}$ has better accuracy than $\widetilde{A}^T \widetilde{B}$ for all possible values of $\theta$, especially when $\theta$ is small.

We now explain the intuition of Eq. (2), and why $\widetilde{M}$ is a better estimator than $\widetilde{A}^T \widetilde{B}$. To estimate the $(i, j)$ entry of $A^T B$, a straightforward way is to use $\widetilde{A}_i^T \widetilde{B}_j = \|\widetilde{A}_i\| \cdot \|\widetilde{B}_j\| \cdot \cos \widetilde{\theta}_{ij}$, where $\widetilde{\theta}_{ij}$ is the

angle between vectors $\widetilde{A}_i$ and $\widetilde{B}_j$. Since we already know the actual column norms, a potentially better estimator would be $\|A_i\| \cdot \|B_j\| \cdot \cos\widetilde{\theta}_{ij}$. This removes the uncertainty that comes from distorted column norms[3].

Figure 2(a) compares the two estimators $\widetilde{A}_i^T\widetilde{B}_j$ (JL embedding) and $\widetilde{M}(i,j)$ (rescaled JL embedding) for dot products. We plot simulation results on pairs of unit-norm vectors with different angles. The vectors have dimension 1,000 and the sketching matrix has dimension 10-by-1,000. Clearly rescaling by the actual norms help reduce the estimation uncertainty. This phenomenon is more prominent when the true dot products are close to $\pm 1$, which makes sense because $\cos\theta$ has a small slope when $\cos\theta$ approaches $\pm 1$, and hence the uncertainty from angles may produce smaller distortion compared to that from norms. In the extreme case when $\cos\theta = \pm 1$, rescaled JL embedding can perfectly recover the true dot product[4].

In the lower part of Figure 2(b) we illustrate how to construct unit-norm vectors from a cone with angle $\theta$. Given a fixed unit-norm vector $x$, and a random Gaussian vector $t$ with expected norm $\tan(\theta/2)$, we construct new vector $y$ by randomly picking one from the two possible choices $x+t$ and $-(x+t)$, and then renormalize it. Suppose the columns of $A$ and $B$ are unit vectors randomly drawn from a cone with angle $\theta$, we plot the ratio of spectral norm errors $\|A^T B - \widetilde{A}^T\widetilde{B}\|/\|A^T B - \widetilde{M}\|$ in Figure 2(b). We observe that $\widetilde{M}$ always outperforms $\widetilde{A}^T\widetilde{B}$ and can be much better when $\theta$ approaches zero, which agrees with the trend indicated in Figure 2(a).

**Step 3: Compute low rank approximation given estimates of few entries of $A^T B$.** Finally we compute the low rank approximation of $A^T B$ from the samples using alternating least squares:

$$\min_{U,V\in\mathbb{R}^{n\times r}} \sum_{(i,j)\in\Omega} w_{ij}(e_i^T UV^T e_j - \widetilde{M}(i,j))^2, \tag{3}$$

where $w_{ij} = 1/\hat{q}_{ij}$ denotes the weights, and $e_i$, $e_j$ are standard base vectors. This is a popular technique for low rank recovery and matrix completion (see [1] and the references therein). After $T$ iterations, we will get a rank-$r$ approximation of $\widetilde{M}$ presented in the convenient factored form. This subroutine is quite standard, so we defer the details to Appendix A.

# 3 Analysis

Now we present the main theoretical result. Theorem 3.1 characterizes the interaction between the sketch size $k$, the sampling complexity $m$, the number of iterations $T$, and the spectral error $\|(A^T B)_r - \widehat{A^T B}_r\|$, where $\widehat{A^T B}_r$ is the output of SMP-PCA, and $(A^T B)_r$ is the optimal rank-$r$ approximation of $A^T B$. Note that the following theorem assumes that $A$ and $B$ have the same size. For the general case of $n_1 \neq n_2$, Theorem 3.1 is still valid by setting $n = \max\{n_1, n_2\}$.

**Theorem 3.1.** *Given matrices $A \in \mathbb{R}^{d\times n}$ and $B \in \mathbb{R}^{d\times n}$, let $(A^T B)_r$ be the optimal rank-$r$ approximation of $A^T B$. Define $\tilde{r} = \max\{\frac{\|A\|_F^2}{\|A\|^2}, \frac{\|B\|_F^2}{\|B\|^2}\}$ as the maximum stable rank, and $\rho = \frac{\sigma_1^*}{\sigma_r^*}$ as the condition number of $(A^T B)_r$, where $\sigma_i^*$ is the $i$-th singular values of $A^T B$.*

*Let $\widehat{A^T B}_r$ be the output of Algorithm SMP-PCA. If the input parameters $k$, $m$, and $T$ satisfy*

$$k \geq \frac{C_1\|A\|^2\|B\|^2\rho^2 r^3}{\|A^T B\|_F^2} \cdot \frac{\max\{\tilde{r}, 2\log(n)\} + \log(3/\gamma)}{\eta^2}, \tag{4}$$

$$m \geq \frac{C_2\tilde{r}^2}{\gamma} \cdot \left(\frac{\|A\|_F^2 + \|B\|_F^2}{\|A^T B\|_F}\right)^2 \cdot \frac{nr^3\rho^2\log(n)T^2}{\eta^2}, \tag{5}$$

$$T \geq \log(\frac{\|A\|_F + \|B\|_F}{\zeta}), \tag{6}$$

*where $C_1$ and $C_2$ are some global constants independent of $A$ and $B$. Then with probability at least $1-\gamma$, we have*

$$\|(A^T B)_r - \widehat{A^T B}_r\| \leq \eta\|A^T B - (A^T B)_r\|_F + \zeta + \eta\sigma_r^*. \tag{7}$$

**Remark 1.** Compared to the two-pass algorithm proposed by [1], we notice that Eq. (7) contains an additional error term $\eta\sigma_r^*$. This extra term captures the cost incurred when we are approximating entries of $A^TB$ by Eq. (2) instead of using the actual values. The exact tradeoff between $\eta$ and $k$ is given by Eq. (4). On one hand, we want to have a small $k$ so that the sketched matrices can fit into memory. On the other hand, the parameter $k$ controls how much information is lost during sketching, and a larger $k$ gives a more accurate estimation of the inner products.

**Remark 2.** The dependence on $\frac{\|A\|_F^2+\|B\|_F^2}{\|A^TB\|_F}$ captures one difficult situation for our algorithm. If $\|A^TB\|_F$ is much smaller than $\|A\|_F$ or $\|B\|_F$, which could happen, e.g., when many column vectors of $A$ are orthogonal to those of $B$, then SMP-PCA requires many samples to work. This is reasonable. Imagine that $A^TB$ is close to an identity matrix, then it may be hard to tell it from an all-zero matrix without enough samples. Nevertheless, removing this dependence is an interesting direction for future research.

**Remark 3.** For the special case of $A = B$, SMP-PCA computes a rank-$r$ approximation of matrix $A^TA$ in a single pass. Theorem 3.1 provides an error bound in spectral norm for the residual matrix $(A^TA)_r - \widehat{A^TA}_r$. Most results in the online PCA literature use Frobenius norm as performance measure. Recently, [10] provides an online PCA algorithm with spectral norm guarantee. They achieves a spectral norm bound of $\epsilon\sigma_1^* + \sigma_{r+1}^*$, which is stronger than ours. However, their algorithm requires a target dimension of $O(r\log n/\epsilon^2)$, i.e., the output is a matrix of size $n$-by-$O(r\log n/\epsilon^2)$, while the output of SMP-PCA is simply $n$-by-$r$.

**Remark 4.** We defer our proofs to Appendix C. The proof proceeds in three steps. In Appendix C.2, we show that the sampled matrix provides a good approximation of the actual matrix $A^TB$. In Appendix C.3, we show that there is a geometric decrease in the distance between the computed subspaces $\widehat{U}, \widehat{V}$ and the optimal ones $U^*, V^*$ at each iteration of WAltMin algorithm. The spectral norm bound in Theorem 3.1 is then proved using results from the previous two steps.

**Computation Complexity.** We now analyze the computation complexity of SMP-PCA. In Step 1, we compute the sketched matrices of $A$ and $B$, which requires $O(\text{nnz}(A)k + \text{nnz}(B)k)$ flops. Here nnz$(\cdot)$ denotes the number of non-zero entries. The main job of Step 2 is to sample a set $\Omega$ and calculate the corresponding inner products, which takes $O(m\log(n) + mk)$ flops. Here we define $n$ as $\max\{n_1, n_2\}$ for simplicity. According to Eq. (4), we have $\log(n) = O(k)$, so Step 2 takes $O(mk)$ flops. In Step 3, we run alternating least squares on the sampled matrix, which can be completed in $O((mr^2 + nr^3)T)$ flops. Since Eq. (5) indicates $nr = O(m)$, the computation complexity of Step 3 is $O(mr^2T)$. Therefore, SMP-PCA has a total computation complexity $O(\text{nnz}(A)k + \text{nnz}(B)k + mk + mr^2T)$.

# 4   Numerical Experiments

**Spark implementation.** We implement our SMP-PCA in Apache Spark 1.6.2 [19]. For the purpose of comparison, we also implement a two-pass algorithm LELA [1] in Spark[5]. The matrices $A$ and $B$ are stored as a resilient distributed dataset (RDD) in disk (by setting its `StorageLevel` as `DISK_ONLY`). We implement the two passes of LELA using the `treeAggregate` method. For SMP-PCA, we choose the subsampled randomized Hadamard transform (SRHT) [16] as the sketching matrix. The biased sampling procedure is performed using binary search (see Appendix C.5 for how to sample $m$ elements in $O(m\log n)$ time). After obtaining the sampled matrix, we run ALS (alternating least squares) to get the desired low-rank matrices. More details can be found at [18].

**Description of datasets.** We test our algorithm on synthetic datasets and three real datasets: SIFT10K [9], NIPS-BW [11], and URL-reputation [12]. For synthetic data, we generate matrices $A$ and $B$ as $GD$, where $G$ has entries independently drawn from standard Gaussian distribution, and $D$ is a diagonal matrix with $D_{ii} = 1/i$. SIFT10K is a dataset of 10,000 images. Each image is represented by 128 features. We set $A$ as the image-by-feature matrix. The task here is to compute a low rank approximation of $A^TA$, which is a standard PCA task. The NIPS-BW dataset contains bag-of-words features extracted from 1,500 NIPS papers. We divide the papers into two subsets, and let $A$ and $B$ be the corresponding word-by-paper matrices, so $A^TB$ computes the counts of co-occurred words between two sets of papers. The original URL-reputation dataset has 2.4 million

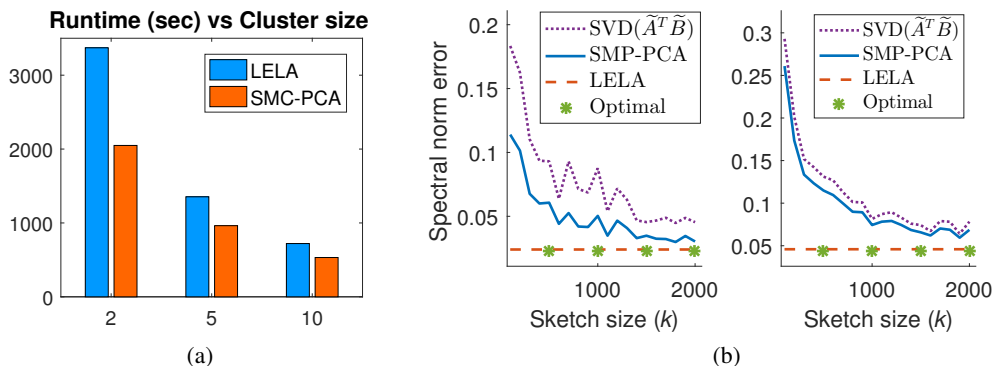

(a)                  (b)

Figure 3: (a) Spark-1.6.2 running time on a 150GB dataset. All nodes are m3.2xlarge EC2 instances. See [18] for more details. (b) Spectral norm error achieved by three algorithms over two datasets: SIFT10K (left) and NIPS-BW (right). SMP-PCA outperforms SVD($\widetilde{A}^T\widetilde{B}$) by a factor of 1.8 for SIFT10K and 1.1 for NIPS-BW. The error of SMP-PCA keeps decreasing as the sketch size $k$ grows.

URLs. Each URL is represented by 3.2 million features, and is indicated as malicious or benign. This dataset has been used previously for CCA [13]. Here we extract two subsets of features, and let $A$ and $B$ be the corresponding URL-by-feature matrices. The goal is to compute a low rank approximation of $A^T B$, the cross-covariance matrix between two subsets of features.

**Sample complexity.** In Figure 4(a) we present simulation results on a small synthetic data with $n = d = 5,000$ and $r = 5$. We observe that a phase transition occurs when the sample complexity $m = \Theta(nr \log n)$. This agrees with the experimental results shown in previous papers, see, e.g., [4, 1]. For all rest experiments, unless otherwise specified, we set $r = 5$, $T = 10$, and $m$ as $4nr \log n$.

Table 1: A comparison of spectral norm error over three datasets

| Dataset | $d$ | $n$ | Algorithm | Sketch size $k$ | Error |
|---|---|---|---|---|---|
| Synthetic | 100,000 | 100,000 | Optimal | - | 0.0271 |
|  |  |  | LELA | - | 0.0274 |
|  |  |  | SMP-PCA | 2,000 | 0.0280 |
| URL-malicious | 792,145 | 10,000 | Optimal | - | 0.0163 |
|  |  |  | LELA | - | 0.0182 |
|  |  |  | SMP-PCA | 2,000 | 0.0188 |
| URL-benign | 1,603,985 | 10,000 | Optimal | - | 0.0103 |
|  |  |  | LELA | - | 0.0105 |
|  |  |  | SMP-PCA | 2,000 | 0.0117 |

**Comparison of SMP-PCA and LELA.** We now compare the statistical performance of SMP-PCA and LELA [1] on three real datasets and one synthetic dataset. As shown in Figure 3(b) and Table 1, LELA always achieves a smaller spectral norm error than SMP-PCA, which makes sense because LELA takes two passes and hence has more chances exploring the data. Besides, we observe that as the sketch size increases, the error of SMP-PCA keeps decreasing and gets closer to that of LELA.

In Figure 3(a) we compare the runtime of SMP-PCA and LELA using a 150GB synthetic dataset on m3.2xlarge Amazon EC2 instances[6]. The matrices $A$ and $B$ have dimension $n = d = 100,000$. The sketch dimension is set as $k = 2,000$. We observe that the speedup achieved by SMP-PCA is more prominent for small clusters (e.g., 56 mins versus 34 mins on a cluster of size two). This is possibly due to the increasing spark overheads at larger clusters, see [8] for more related discussion.

**Comparison of SMP-PCA and SVD($\widetilde{A}^T\widetilde{B}$).** In Figure 4(b) we repeat the experiment in Section 2 by generating column vectors of $A$ and $B$ from a cone with angle $\theta$. Here SVD($\widetilde{A}^T\widetilde{B}$) refers to

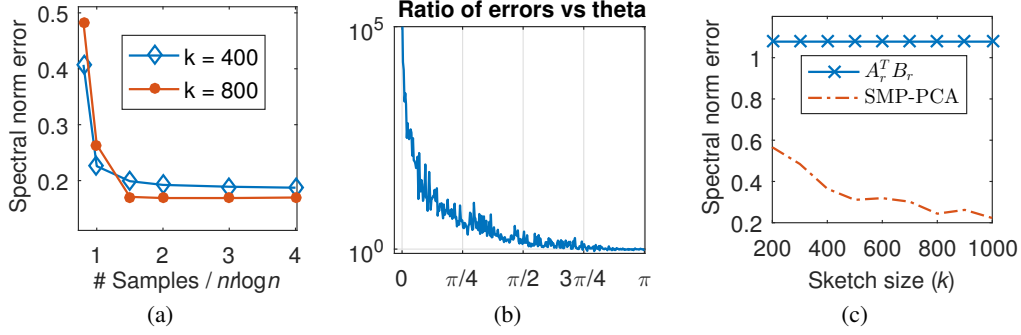

Figure 4: (a) A phase transition occurs when the sample complexity $m = \Theta(nr \log n)$. (b) This figure plots the ratio of spectral norm error of $\text{SVD}(\widetilde{A}^T \widetilde{B})$ over that of SMP-PCA. The columns of $A$ and $B$ are unit vectors drawn from a cone with angle $\theta$. We see that the ratio of errors scales to infinity as the cone angle shrinks. (c) If the top $r$ left singular vectors of $A$ are orthogonal to those of $B$, the product $A_r^T B_r$ is a very poor low rank approximation of $A^T B$.

computing SVD on the sketched matrices[7]. We plot the ratio of the spectral norm error of $\text{SVD}(\widetilde{A}^T \widetilde{B})$ over that of SMP-PCA, as a function of $\theta$. Note that this is different from Figure 2(b), as now we take the effect of random sampling and SVD into account. However, the trend in both figures are the same: SMP-PCA always outperforms $\text{SVD}(\widetilde{A}^T \widetilde{B})$ and can be arbitrarily better as $\theta$ goes to zero.

In Figure 3(b) we compare SMP-PCA and $\text{SVD}(\widetilde{A}^T \widetilde{B})$ on two real datasets SIFK10K and NIPS-BW. The y-axis represents spectral norm error, defined as $||A^T B - \widehat{A^T B}_r||/||A^T B||$, where $\widehat{A^T B}_r$ is the rank-$r$ approximation found by a specific algorithm. We observe that SMP-PCA outperforms $\text{SVD}(\widetilde{A}^T \widetilde{B})$ by a factor of 1.8 for SIFT10K and 1.1 for NIPS-BW.

Now we explain why SMP-PCA produces a more accurate result than $\text{SVD}(\widetilde{A}^T \widetilde{B})$. The reasons are twofold. First, our rescaled JL embedding $\widetilde{M}$ is a better estimator for $A^T B$ than $\widetilde{A}^T \widetilde{B}$ (Figure 2). Second, the noise due to sampling is relatively small compared to the benefit obtained from $\widetilde{M}$, and hence the final result computed using $P_\Omega(\widetilde{M})$ still outperforms $\text{SVD}(\widetilde{A}^T \widetilde{B})$.

**Comparison of SMP-PCA and $A_r^T B_r$.** Let $A_r$ and $B_r$ be the optimal rank-$r$ approximation of $A$ and $B$, we show that even if one could use existing methods (e.g., algorithms for streaming PCA) to estimate $A_r$ and $B_r$, their product $A_r^T B_r$ can be a very poor low rank approximation of $A^T B$. This is demonstrated in Figure 4(c), where we intentionally make the top $r$ left singular vectors of $A$ orthogonal to those of $B$.

## 5   Conclusion

We develop a novel one-pass algorithm SMP-PCA that directly computes a low rank approximation of matrix product, using ideas of matrix sketching and entrywise sampling. As a subroutine of our algorithm, we propose rescaled JL for estimating entries of $A^T B$, which has smaller error compared to the standard estimator $\tilde{A}^T \tilde{B}$. This we believe can be extended to other applications. Moreover, SMP-PCA allows the non-zero entries of $A$ and $B$ to be presented in any arbitrary order, and hence can be used for steaming applications. We design a distributed implementation for SMP-PCA. Our experimental results show that SMP-PCA can perform arbitrarily better than $\text{SVD}(\widetilde{A}^T \widetilde{B})$, and is significantly faster compared to algorithms that require two or more passes over the data.

**Acknowledgements** We thank the anonymous reviewers for their valuable comments. This research has been supported by NSF Grants CCF 1344179, 1344364, 1407278, 1422549, 1302435, 1564000, and ARO YIP W911NF-14-1-0258.

## Footnotes

[1]The code can be found at https://github.com/wushanshan/MatrixProductPCA

[2]One straightforward idea is to sketch each matrix individually and perform SVD on the product of the sketches. We compare against that scheme and show that we can perform arbitrarily better using our rescaled JL embedding.

[3]We also tried using the cosine rule for computing the dot product, and another sketching method specifically designed for preserving angles [2], but empirically those methods perform worse than our current estimator.

[4]See http://wushanshan.github.io/files/RescaledJL_project.pdf for more results.

[5]To our best knowledge, this the first distributed implementation of LELA.

[6]Each machine has 8 cores, 30GB memory, and 2×80GB SSD.

[7]This can be done by standard power iteration based method, without explicitly forming the product matrix $\widetilde{A}^T \widetilde{B}$, whose size is too big to fit into memory according to our assumption.

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
