[Supplementary Material · jl_lowrank_supp.pdf]

# A  Weighted alternating minimization

Algorithm 2 provides a detailed explanation of WAltMin, which follows a standard procedure for matrix completion. We use $R_\Omega(A) = w. * P_\Omega(A)$ to denote the Hadamard product between $w$ and $P_\Omega(A)$: $R_\Omega(A)(i,j) = w(i,j) * P_\Omega(A)(i,j)$ for $(i,j) \in \Omega$ and 0 otherwise, where $w \in \mathbb{R}^{n_1 \times n_2} = 1/\hat{q}_{ij}$ is the weight matrix. Similarly we define the matrix $R_\Omega^{1/2}(A)$ as $R_\Omega^{1/2}(A)(i,j) = \sqrt{w(i,j)} * P_\Omega(A)(i,j)$ for $(i,j) \in \Omega$ and 0 otherwise.

The algorithm contains two parts: initialization (Step 2-6) and weighted alternating minimization (Step 7-10). In the first part, we compute SVD of the weighted sampled matrix $R_\Omega(\widetilde{M})$ and then set row $i$ of $U^{(0)}$ to be zero if its norm is larger than a threshold (Step 6). More details of this trim step can be found in [3]. In the second part, the goal is to solve the following non-convex problem by alternating minimization:

$$\min_{U,V} \sum_{(i,j)\in\Omega} w_{ij}(e_i^T U V^T e_j - \widetilde{M}(i,j))^2, \tag{8}$$

where $e_i$, $e_j$ are standard base vectors. After running $T$ iterations, the algorithm outputs a rank-$r$ approximation of $\widetilde{M}$ presented in the convenient factored form.

---

**Algorithm 2** WAltMin [3]

---

1: **Input:** $P_\Omega(\widetilde{M}) \in \mathbb{R}^{n_1 \times n_2}$, $\Omega$, $r$, $\hat{q}$, and $T$
2: $w_{ij} = 1/\hat{q}_{ij}$ when $\hat{q}_{ij} > 0$, 0 else, $\forall i, j$
3: Divide $\Omega$ in $2T + 1$ equal uniformly random subsets, i.e., $\Omega = \{\Omega_0, \dots, \Omega_{2T}\}$
4: $R_{\Omega_0}(\widetilde{M}) = w. * P_{\Omega_0}(\widetilde{M})$
5: $U^{(0)}\Sigma^{(0)}(V^{(0)})^T = \text{SVD}(R_{\Omega_0}(\widetilde{M}), r)$
6: Trim $U^{(0)}$ and let $\widehat{U}^{(0)}$ be the output
7: **for** $t = 0$ to $T - 1$ **do**
8: $\quad \widehat{V}^{(t+1)} = \text{argmin}_V \|R_{\Omega_{2t+1}}^{1/2}(\widetilde{M} - \widehat{U}^{(t)}V^T)\|_F^2$
9: $\quad \widehat{U}^{(t+1)} = \text{argmin}_U \|R_{\Omega_{2t+2}}^{1/2}(\widetilde{M} - U(\widehat{V}^{(t+1)})^T)\|_F^2$
10: **end for**
11: **Output:** $\widehat{U}^{(T)} \in \mathbb{R}^{n_1 \times r}$ and $\widehat{V}^{(T)} \in \mathbb{R}^{n_2 \times r}$.

---

# B  Technical Lemmas

We will frequently use the following concentration inequality in the proof.

**Lemma B.1.** *(Matrix Bernstein's Inequality [33]).  Consider $p$ independent random matrices $X_1, ...., X_p$ in $\mathbb{R}^{n \times n}$, where each matrix has bounded deviation from its mean:*

$$\|X_i - \mathbb{E}[X_i]\| \leq L, \quad \forall i.$$

*Let the norm of the covariance matrix be*

$$\sigma^2 = \max\left\{\left\|\mathbb{E}\left[\sum_{i=1}^p (X_i - \mathbb{E}[X_i])(X_i - \mathbb{E}[X_i])^T\right]\right\|, \left\|\mathbb{E}\left[\sum_{i=1}^p (X_i - \mathbb{E}[X_i])^T(X_i - \mathbb{E}[X_i])\right]\right\|\right\}$$

*Then the following holds for all $t \geq 0$:*

$$\mathbb{P}\left[\left\|\sum_{i=1}^p (X_i - \mathbb{E}[X_i])\right\|\right] \leq 2n \exp(\frac{-t^2/2}{\sigma^2 + Lt/3}).$$

A formal definition of JL transform is given below [29][34].

**Definition B.2.** *A random matrix $\Pi \in \mathbb{R}^{k \times d}$ forms a JL transform with parameters $\epsilon, \delta, f$ or JLT($\epsilon, \delta, f$) for short, if with probability at least $1 - \delta$, for any $f$-element subset $V \subset \mathbb{R}^d$, for all $v, v' \in V$ it holds that $|\langle \Pi v, \Pi v' \rangle - \langle v, v' \rangle| \leq \epsilon \|v\| \cdot \|v'\|$.*

The following lemma [34] characterizes the tradeoff between the reduced dimension $k$ and the error level $\epsilon$.

**Lemma B.3.** *Let $0 < \epsilon, \delta < 1$, and $\Pi \in \mathbb{R}^{k \times d}$ be a random matrix where the entries $\Pi(i,j)$ are i.i.d. $\mathcal{N}(0, 1/k)$ random variables. If $k = \Omega(\log(f/\delta)\epsilon^{-2})$, then $\Pi$ is a JLT($\epsilon, \delta, f$).*

We now present two lemmas that connect $\widetilde{A} \in \mathbb{R}^{k \times n}$ and $\widetilde{B} \in \mathbb{R}^{d \times n}$ with $A \in \mathbb{R}^{d \times n}$ and $B \in \mathbb{R}^{d \times n}$.

**Lemma B.4.** *Let $0 < \epsilon, \delta < 1$, if $k = \Omega(\frac{\log(2n/\delta)}{\epsilon^2})$, then with probability at least $1 - \delta$,*

$$(1 - \epsilon)||A||_F^2 \leq ||\widetilde{A}||_F^2 \leq (1 + \epsilon)||A||_F^2, \quad (1 - \epsilon)||B||_F^2 \leq ||\widetilde{B}||_F^2 \leq (1 + \epsilon)||B||_F^2,$$

$$||\widetilde{A}^T \widetilde{B} - A^T B||_F \leq \epsilon ||A||_F ||B||_F.$$

*Proof.* This is again a standard result of JL transformation, e.g., see Definition 2.3 and Theorem 2.1 of [34] and Lemma 6 of [29] . $\square$

**Lemma B.5.** *Let $0 < \epsilon, \delta < 1$, if $k = \Theta(\frac{\tilde{r} + \log(1/\delta)}{\epsilon^2})$, where $\tilde{r} = \max\{\frac{||A||_F^2}{||A||^2}, \frac{||B||_F^2}{||B||^2}\}$ is the maximum stable rank, then with probability at least $1 - \delta$,*

$$||\widetilde{A}^T \widetilde{B} - A^T B|| \leq \epsilon ||A|| ||B||.$$

*Proof.* This follows from a recent paper [12]. $\square$

Using the above two lemmas, we can prove the following two lemmas that relate $\widetilde{M}$ with $A^T B$, for $\widetilde{M}$ defined in Algorithm 1. A more compact definition of $\widetilde{M}$ is $D_A \widetilde{A}^T \widetilde{B} D_B$, where $D_A$ and $D_B$ are diagonal matrices with $(D_A)_{ii} = ||A_i||/||\widetilde{A}_i||$ and $(D_B)_{jj} = ||B_j||/||\widetilde{B}_j||$.

**Lemma B.6.** *Let $0 < \epsilon < 1/14$, $0 < \delta < 1$, if $k = \Omega(\frac{\log(2n/\delta)}{\epsilon^2})$, then with probability at least $1 - \delta$,*

$$|\widetilde{M}_{ij} - A_i^T B_j| \leq \epsilon ||A_i|| \cdot ||B_j||, \quad ||\widetilde{M} - A^T B||_F \leq \epsilon ||A||_F ||B||_F.$$

*Proof.* Let $0 < \epsilon < 1/2$, $0 < \delta < 1$, according to the Definition B.2 and Lemma B.3, we have that if $k = \Omega(\frac{\log(2n/\delta)}{\epsilon^2})$, then with probability at least $1 - \delta$, and for all $i, j$

$$1 - \epsilon \leq (D_A)_{ii} \leq 1 + \epsilon, \quad 1 - \epsilon \leq (D_B)_{jj} \leq 1 + \epsilon, \quad |\widetilde{A}_i^T \widetilde{B}_j - A_i^T B_j| \leq \epsilon ||A_i|| ||B_j||. \quad (9)$$

We can now bound $|\widetilde{M}_{ij} - A_i^T B_j|$ as

$$|\widetilde{M}_{ij} - A_i^T B_j|$$
$$\overset{\xi_1}{=} |\widetilde{A}_i^T \widetilde{B}_j (D_A)_{ii} (D_B)_{jj} - A_i^T B_j|$$
$$\overset{\xi_2}{\leq} \max\{|\widetilde{A}_i^T \widetilde{B}_j (1 + \epsilon)^2 - A_i^T B_j|, |\widetilde{A}_i^T \widetilde{B}_j (1 - \epsilon)^2 - A_i^T B_j|\}$$
$$\overset{\xi_3}{\leq} \max\{(1 + \epsilon)^2 \epsilon ||A_i|| ||B_j|| + ((1 + \epsilon)^2 - 1)|A_i^T B_j|, (1 - \epsilon)^2 \epsilon ||A_i|| ||B_j|| + (1 - (1 - \epsilon)^2)|A_i^T B_j|\}$$
$$\overset{\xi_4}{\leq} 7\epsilon ||A_i|| ||B_j||, \tag{10}$$

where $\xi_1$ follows from the definition of $\widetilde{M}_{ij}$, $\xi_2$ follows from the bound in Eq.(9), $\xi_3$ follows from triangle inequality and Eq.(9), and $\xi_4$ follows from $|A_i^T B_j| \leq ||A_i|| ||B_j||$. Now rescaling $\epsilon$ as $\epsilon/7$ gives the desired bound in Lemma B.6.

Hence, $||\widetilde{M} - A^T B||_F = \sqrt{\sum_{ij} |\widetilde{M}_{ij} - A_i^T B_j|^2} \leq \sqrt{\sum_{ij} \epsilon^2 ||A_i||^2 ||B_j||^2} = \epsilon ||A||_F ||B||_F$. $\square$

**Lemma B.7.** *Let $0 < \epsilon < 1/14$, $0 < \delta < 1$, if $k = \Omega(\frac{\tilde{r} + \log(n/\delta)}{\epsilon^2})$, then with probability at least $1 - \delta$,*

$$||\widetilde{M} - A^T B|| \leq \epsilon ||A|| ||B||.$$

*Proof.* We can bound the spectral norm of the difference matrix as follows:

$$||\widetilde{M} - A^T B|| \overset{\xi_1}{=} ||D_A \widetilde{A}^T \widetilde{B} D_B - D_A A^T B D_B + D_A A^T B D_B - D_A A^T B + D_A A^T B - A^T B||$$

$$\leq ||D_A||\,||\widetilde{A}^T \widetilde{B} - A^T B||\,||D_B|| + ||D_A||\,||A^T B||\,||D_B - I|| + ||D_A - I||\,||A^T B||$$

$$\overset{\xi_3}{\leq} (1+\epsilon)^2 \epsilon ||A||\,||B|| + (1+\epsilon)\epsilon ||A||\,||B|| + \epsilon ||A||\,||B||$$

$$\leq 7\epsilon ||A||\,||B||, \tag{11}$$

where $\xi_1$ follows from the definition of $\widetilde{M}_{ij}$, and $\xi_2$ follows from Lemma B.5 and bound in Eq.(9). Rescaling $\epsilon$ as $\epsilon/7$ gives the desired bound in Lemma B.7. □

We will frequently use the term *with high probability*. Here is a formal definition.

**Definition B.8.** *We say that an event $E$ occurs with high probability (w.h.p.) in $n$ if the probability that its complement $\bar{E}$ happens is polynomially small, i.e., $Pr(\bar{E}) = O(\frac{1}{n^\alpha})$ for some constant $\alpha > 0$.*

The following two lemmas define a "nice" $\Pi$ and when this happens with high probability.

**Definition B.9.** *The random Gaussian matrix $\Pi$ is "nice" with parameter $\epsilon$ if for all $(i,j)$ such that $q_{ij} \leq 1$ (i.e., $q_{ij} = \hat{q}_{ij}$), the sketched values $\widetilde{M}_{ij}$ satisfies the following two inequalities:*

$$\frac{|\widetilde{M}_{ij}|}{\hat{q}_{ij}} \leq (1+\epsilon)\frac{n}{m}(||A||_F^2 + ||B||_F^2), \qquad \sum_{\{j:\hat{q}_{ij}=q_{ij}\}} \frac{\widetilde{M}_{ij}^2}{\hat{q}_{ij}} \leq (1+\epsilon)\frac{2n}{m}(||A||_F^2 + ||B||_F^2)^2.$$

**Lemma B.10.** *If $k = \Omega(\frac{\log(n)}{\epsilon^2})$, and $0 < \epsilon < 1/14$, then the random Gaussian matrix $\Pi \in \mathbb{R}^{k \times d}$ is "nice" w.h.p. in $n$.*

*Proof.* According to Lemma B.6, if $k = \Omega(\frac{\log(n)}{\epsilon^2})$, then w.h.p. in $n$, for all $(i,j)$ we have $|\widetilde{M}_{ij} - A_i^T B_j| \leq \epsilon ||A_i|| \cdot ||B_j||$. In other words, the following holds with probability at least $1 - \delta$:

$$|\widetilde{M}_{ij}| \leq |A_i^T B_j| + \epsilon ||A_i|| \cdot ||B_j|| \leq (1+\epsilon)||A_i|| \cdot ||B_j||, \quad \forall(i,j)$$

The above inequality is sufficient for $\Pi$ to be "nice":

$$\frac{\widetilde{M}_{ij}}{\hat{q}_{ij}} \leq (1+\epsilon)\frac{||A_i|| \cdot ||B_j||}{\hat{q}_{ij}} \leq (1+\epsilon)\frac{(||A_i||^2 + ||B_j||^2)/2}{m \cdot (\frac{||A_i||^2}{2n||A||_F^2} + \frac{||B_j||^2}{2n||B||_F^2})} \leq (1+\epsilon)\frac{n}{m}(||A||_F^2 + ||B||_F^2)$$

$$\sum_{\{j:\hat{q}_{ij}=q_{ij}\}} \frac{\widetilde{M}_{ij}^2}{\hat{q}_{ij}} \leq \sum_{\{j:\hat{q}_{ij}=q_{ij}\}} \frac{(1+\epsilon)^2||A_i||^2||B_j||^2}{\hat{q}_{ij}}$$

$$\leq (1+\epsilon) \sum_{\{j:\hat{q}_{ij}=q_{ij}\}} \frac{||A_i||^4 + ||B_j||^4}{m \cdot (\frac{||A_i||^2}{2n||A||_F^2} + \frac{||B_j||^2}{2n||B||_F^2})}$$

$$\leq (1+\epsilon)\frac{2n}{m}(||A||_F^2 + ||B||_F^2)^2.$$

Therefore, we conclude that if $k = \Omega(\frac{\log(n)}{\epsilon^2})$, then $\Pi$ is "nice" w.h.p. in $n$. □

## C  Proofs

### C.1  Proof overview

We now present the key steps in proving Theorem 3.1. The framework is similar to that of LELA [3].

Our proof proceeds in three steps. In the first step, we show that the sampled matrix provides a good approximation of the actual matrix $A^T B$. The result is summarized in Lemma C.1. Here $R_\Omega(\widetilde{M})$ denotes the sampled matrix weighted by the inverse of sampling probability (see Line 4 of Algorithm 2). Detailed proof can be found in Appendix C.2. For consistency, we will use $C_i$ $(i = 1, 2, ...)$ to denote global constant that can vary from step to step.

**Lemma C.1.** (Initialization) *Let $m$ and $k$ satisfy the following conditions for sufficiently large constants $C_1$ and $C_2$:*

$$m \geq C_1 \left( \frac{||A||_F^2 + ||B||_F^2}{||A^T B||_F} \right)^2 \frac{n}{\delta^2} \log(n),$$

$$k \geq C_2 \frac{\tilde{r} + \log(n)}{\delta^2} \cdot \frac{||A||^2 ||B||^2}{||A^T B||_F^2},$$

*then the following holds w.h.p. in $n$:*

$$||R_\Omega(\widetilde{M}) - A^T B|| \leq \delta ||A^T B||_F.$$

In the second step, we show that at each iteration of WAltMin algorithm, there is a geometric decrease in the distance between the computed subspaces $\widehat{U}$, $\widehat{V}$ and the optimal ones $U^*$, $V^*$. The result is shown in Lemma C.2. Appendix C.3 provides the detailed proof. Here for any two orthonormal matrices $X$ and $Y$, we define their distance as the principal angle based distance, i.e., $dist(X, Y) = ||X_\perp^T Y||$, where $X_\perp$ denotes the subspace orthogonal to $X$.

**Lemma C.2.** (WAltMin Descent) *Let $k$, $m$, and $T$ satisfy the conditions stated in Theorem 3.1. Also, consider the case when $||A^T B - (A^T B)_r||_F \leq \frac{1}{576\rho r^{1.5}} ||(A^T B)_r||_F$. Let $\hat{U}^{(t)}$ and $\hat{V}^{(t+1)}$ be the $t$-th and $(t+1)$-th step iterates of the WAltMin procedure. Let $U^{(t)}$ and $V^{(t+1)}$ be the corresponding orthonormal matrices. Let $||(U^{(t)})^i|| \leq 8\sqrt{r}\rho ||A_i||/||A||_F$ and $dist(U^{(t)}, U^*) \leq 1/2$. Denote $A^T B$ as $M$, then the following holds with probability at least $1 - \gamma/T$:*

$$dist(V^{t+1}, V^*) \leq \frac{1}{2} dist(U^t, U^*) + \eta ||M - M_r||_F / \sigma_r^* + \eta,$$

$$||(V^{(t+1)})^j|| \leq 8\sqrt{r}\rho ||B_j||/||B||_F.$$

In the third step, we prove the spectral norm bound in Theorem 3.1 using results from the above two lemmas. Comparing Lemma C.1 and C.2 with their counterparts of LELA (see Lemma C.2 and C.3 in [3]), we notice that Lemma C.1 has the same bound as that of LELA, but the bound in Lemma C.2 contains an extra term $\eta$. This term eventually leads to an additive error term $\eta\sigma_r^*$ in Eq.(7). Detailed proof is in Appendix C.4.

## C.2 Proof of Lemma C.1

We first prove the following lemma, which shows that $R_\Omega(\widetilde{M})$ is close to $\widetilde{M}$. For simplicity of presentation, we define $C_{AB} := \frac{(||A||_F^2 + ||B||_F^2)^2}{||A^T B||_F^2}$.

**Lemma C.3.** *Suppose $\Pi$ is fixed and is "nice". Let $m \geq C_1 \cdot C_{AB} \frac{n}{\delta^2} \log(n)$ for sufficiently large global constant $C_1$, then w.h.p. in $n$, the following is true:*

$$||R_\Omega(\widetilde{M}) - \widetilde{M}|| \leq \delta ||A^T B||_F.$$

*Proof.* This lemma can be proved in the same way as the proof of Lemma C.2 in [3]. The key idea is to use the matrix Bernstein inequality. Let $X_{ij} = (\delta_{ij} - \hat{q}_{ij}) w_{ij} \widetilde{M}_{ij} e_i e_j^T$, where $\delta_{ij}$ is a $\{0, 1\}$ random variable indicating whether the value at $(i, j)$ has been sampled. Since $\Pi$ is fixed, $\{X_{ij}\}_{i,j=1}^n$ are independent zero mean random matrices. Furthermore, $\sum_{i,j} \{X_{ij}\}_{i,j=1}^n = R_\Omega(\widetilde{M}) - \widetilde{M}$.

Since $\Pi$ is "nice" with parameter $0 < \epsilon < 1/14$, we can bound the 1st and 2nd moment of $X_{ij}$ as follows:

$$||X_{ij}|| = \max\{|(1 - \hat{q}_{ij}) w_{ij} \widetilde{M}_{ij}|, |\hat{q}_{ij} w_{ij} \widetilde{M}_{ij}|\} \leq \frac{|\widetilde{M}_{ij}|}{\hat{q}_{ij}} \overset{\xi_1}{\leq} (1 + \epsilon) \frac{n}{m} (||A||_F^2 + ||B||_F^2);$$

$$\sigma^2 = \max\{\left\|\mathbb{E}\left[\sum_{ij} X_{ij} X_{ij}^T\right]\right\|, \left\|\mathbb{E}\left[\sum_{ij} X_{ij}^T X_{ij}\right]\right\|\} \overset{\xi_2}{=} \max_i \left|\sum_j \hat{q}_{ij}(1 - \hat{q}_{ij}) w_{ij}^2 \widetilde{M}_{ij}^2\right|$$

$$= \max_i |(\frac{1}{\hat{q}_{ij}} - 1)\widetilde{M}_{ij}^2| \overset{\xi_3}{\leq} \sum_{\{j: \hat{q}_{ij} = q_{ij}\}} \frac{\widetilde{M}_{ij}^2}{\hat{q}_{ij}} \overset{\xi_1}{\leq} (1 + \epsilon) \frac{2n}{m} (||A||_F^2 + ||B||_F^2)^2,$$

where $\xi_1$ follows from Lemma B.10, $\xi_2$ follows from a direct calculation, and $\xi_3$ follows from the fact that $\hat{q}_{ij} \leq 1$. Now we can use matrix Bernstein inequality (see Lemma B.1) with $t = \delta ||A^T B||_F$ to show that if $m \geq (1 + \epsilon) C_1 C_{AB} \frac{n}{\delta^2} \log(n)$, then the desired inequality holds w.h.p. in $n$, where $C_1$ is some global constant independent of $A$ and $B$. Note that since $0 < \epsilon < 1/14$, $(1 + \epsilon) < 2$. Rescaling $C_1$ gives the desired result. $\square$

Now we are ready to **prove Lemma C.1**, which is a counterpart of Lemma C.2 in [3].

*Proof.* We first show that $||R_\Omega(\widetilde{M}) - \widetilde{M}|| \leq \delta ||A^T B||_F$ holds w.h.p. in $n$ over the randomness of $\Pi$. Note that in Lemma C.3, we have shown that it is true for a fixed and "nice" $\Pi$, now we want to show that it also holds w.h.p. in $n$ even for a random chosen $\Pi$.

Let $G$ be the event that we desire, i.e., $G = \{||R_\Omega(\widetilde{A}^T \widetilde{B}) - \widetilde{A}^T \widetilde{B}|| \leq \delta ||A^T B||_F\}$. Let $\bar{G}$ be the complimentary event. By conditioning on $\Pi$, we can bound the probability of $\bar{G}$ as

$$Pr(\bar{G}) = Pr(\bar{G}|\Pi \text{ is "nice"})Pr(\Pi \text{ is "nice"}) + Pr(\bar{G}|\Pi \text{ is not "nice"})Pr(\Pi \text{ is not "nice"})$$
$$\leq Pr(\bar{G}|\Pi \text{ is "nice"}) + Pr(\Pi \text{ is not "nice"}).$$

According to Lemma C.3 and Lemma B.10, if $m \geq C_1 \cdot C_{AB} \frac{n}{\delta^2} \log(n)$, and $k \geq C_2 \frac{\log(n)}{\epsilon^2}$, then both events $\{G|\Pi \text{ is "nice"}\}$ and $Pr(\Pi \text{ is "nice"})$ happen w.h.p. in $n$. Therefore, the the probability of $\bar{G}$ is polynomially small in $n$, i.e., the desired event $G$ happens w.h.p. in $n$.

Next we show that $||\widetilde{M} - A^T B|| \leq \delta ||A^T B||_F$ holds w.h.p. in $n$. According to Lemma B.7, if $k = \Theta(\frac{\tilde{r} + \log(n)}{\epsilon^2})$, then w.h.p. in $n$, we have $||\widetilde{M} - A^T B|| \leq \epsilon ||A|| ||B||$. Now let $\epsilon := \delta \frac{||A^T B||_F}{||A|| ||B||}$, we have that if $k = \Theta(\frac{\tilde{r} + \log(n)}{\delta^2} \cdot \frac{||A||^2 ||B||^2}{||A^T B||_F^2})$, then $||\widetilde{M} - A^T B|| \leq \delta ||A^T B||_F$ holds w.h.p. in $n$.

By triangle inequality, we have $||R_\Omega(\widetilde{M}) - A^T B|| \leq ||R_\Omega(\widetilde{M}) - \widetilde{M}|| + ||\widetilde{M} - A^T B||$. We have shown that w.h.p. in $n$, both terms are less than $\delta ||A^T B||_F$. By rescaling $\delta$ as $\delta/2$, we have that the desired inequality $||R_\Omega(\widetilde{A}^T \widetilde{B}) - A^T B|| \leq \delta ||A^T B||_F$ holds w.h.p. in $n$, when $m$ and $k$ are chosen according to the statement of Lemma C.1. $\square$

Because the bound of Lemma C.1 has the same form as that of Lemma C.2 in [3], the corollary of Lemma C.2 also holds for $R_\Omega(\widetilde{M})$, which is stated here without proof: if $||A^T B - (A^T B)_r||_F \leq \frac{1}{576\kappa r^{1.5}} ||(A^T B)_r||_F$, then w.h.p. in $n$ we have

$$||(\widehat{U}^{(0)})^i|| \leq 8\sqrt{r} ||A_i||/||A||_F \quad \text{and} \quad dist(\widehat{U}^{(0)}, U^*) \leq 1/2,$$

where $\widehat{U}^{(0)}$ is the initial iterate produced by the WAltMin algorithm (see Step 6 of Algorithm 2). This corollary will be used in the proof of Lemma C.2.

Similar to the original proof in [3], we can now consider two cases separately: (1) $||A^T B - (A^T B)_r||_F \geq \frac{1}{576\rho r^{1.5}} ||(A^T B)_r||_F$; (2) $||A^T B - (A^T B)_r||_F \leq \frac{1}{576\rho r^{1.5}} ||(A^T B)_r||_F$. The first case is simple: use Lemma C.1 and Wely's inequality [31] already implies the desired bound in Theorem 3.1. To see why, note that Lemma C.1 and Wely's inequality imply that

$$||(A^T B)_r - (R_\Omega(\widetilde{M})_r)||$$
$$\overset{\xi_1}{\leq} ||A^T B - (A^T B)_r|| + ||A^T B - R_\Omega(\widetilde{M})|| + ||R_\Omega(\widetilde{M}) - (R_\Omega(\widetilde{M}))_r||$$
$$\overset{\xi_2}{\leq} ||A^T B - (A^T B)_r|| + \delta ||A^T B||_F + ||R_\Omega(\widetilde{M}) - A^T B|| + ||A^T B - (A^T B)_r||$$
$$\overset{\xi_3}{\leq} 2||A^T B - (A^T B)_r|| + 2\delta ||A^T B||_F, \tag{12}$$

where $M_r$ denotes the best rank-$r$ approximation of $M$, $\xi_1$ follows triangle inequality, $\xi_2$ follows from Lemma C.1 and Wely's inequality, and $\xi_3$ follows from Lemma C.1. If $||A^T B - (A^T B)_r||_F \geq \frac{1}{576\rho r^{1.5}} ||(A^T B)_r||_F$, then $||A^T B||_F = ||(A^T B)_r||_F + ||A^T B - (A^T B)_r||_F \leq O(\rho r^{1.5})||A^T B - (A^T B)_r||_F$. Setting $\delta = O(\eta/(\rho r^{1.5}))$ in Eq.(12) gives the desired error bound in Theorem 3.1. Therefore, in the following analysis we only need to consider the second case.

## C.3 Proof of Lemma C.2

We first prove the following lemma, which is a counterpart of Lemma C.5 in [3]. For simplicity of presentation, we use $M$ to denote $A^T B$ in the following proof.

**Lemma C.4.** *If $m \geq C_1 nr \log(n) T/(\gamma\delta^2)$ and $k \geq C_2(\tilde{r} + \log(n))/\epsilon^2$ for sufficiently large global constants $C_1$ and $C_2$, then the following holds with probability at least $1 - \gamma/T$:*

$$||(U^{(t)})^T R_\Omega(\widetilde{M} - M_r) - (U^{(t)})^T(M - M_r)|| \leq \delta||M - M_r||_F + \delta\epsilon||A||_F||B||_F + \epsilon||A||||B||.$$

*Proof.* For a fixed $\Pi$, we have that if $m \geq C_1 nr \log(n) T/(\gamma\delta^2)$, then following holds with probability at least $1 - \gamma/T$:

$$||(U^{(t)})^T R_\Omega(\widetilde{M} - M_r) - (U^{(t)})^T(\widetilde{M} - M_r)|| \leq \delta||\widetilde{M} - M_r||_F. \tag{13}$$

The proof of Eq.(13) is exactly the same as the proof of Lemma C.5/B.6/B.2 in [3], so we omit its details here. The key idea is to define a set of zero-mean random matrices $X_{ij}$ such that $\sum_{ij} X_{ij} = (U^{(t)})^T R_\Omega(\widetilde{M} - M_r) - (U^{(t)})^T(\widetilde{M} - M_r)$, and then use second moment-based matrix Chebyshev inequality to obtain the desired bound.

According to Lemma B.6 and Lemma B.7, if $k = \Theta((\tilde{r} + \log(n))/\epsilon^2)$, then w.h.p. in $n$, the following holds:

$$||\widetilde{M} - A^T B||_F \leq \epsilon||A||_F||B||_F, \quad ||\widetilde{M} - A^T B|| \leq \epsilon||A||||B||. \tag{14}$$

Using triangle inequality, we have that if $m$ and $k$ satisfy the conditions of Lemma C.4, then the following holds with probability at least $1 - \gamma/T$:

$$
\begin{aligned}
&||(U^{(t)})^T R_\Omega(\widetilde{M} - M_r) - (U^{(t)})^T(M - M_r)|| \\
&\leq ||(U^{(t)})^T R_\Omega(\widetilde{M} - M_r) - (U^{(t)})^T(\widetilde{M} - M_r)|| + ||(U^{(t)})^T(M - \widetilde{M})|| \\
&\overset{\xi_1}{\leq} \delta||\widetilde{M} - M_r||_F + ||M - \widetilde{M}|| \\
&\leq \delta||M - M_r||_F + \delta||M - \widetilde{M}||_F + ||M - \widetilde{M}|| \\
&\overset{\xi_2}{\leq} \delta||M - M_r||_F + \delta\epsilon||A||_F||B||_F + \epsilon||A||||B||,
\end{aligned}
$$

where $\xi_1$ follows from Eq.(13), and $\xi_2$ follows from Eq.(14). $\qquad\square$

Now we are ready to **prove Lemma C.2**. For simplicity, we focus on the rank-1 case here. Rank-$r$ proof follows a similar line of reasoning and can be obtained by combining the current proof with the rank-$r$ analysis in the original proof of LELA [3]. Note that compared to Lemma C.5 in [3], Lemma C.4 contains two extra terms $\delta\epsilon||A||_F||B||_F + \epsilon||A||||B||$. Therefore, we need to be careful for steps that involve Lemma C.4.

In the rank-1 case, we use $\hat{u}^t$ and $\hat{v}^{t+1}$ to denote the $t$-th and $(t+1)$-th step iterates (which are column vectors in this case) of the WAltMin algorithm. Let $u^t$ and $v^{t+1}$ be the corresponding normalized vectors.

*Proof.* This proof contains two parts. In the first part, we will prove that the distance $dist(v^{t+1}, v^*)$ decreases geometrically over time. In the second part, we show that the $j$-th entry of $v^{t+1}$ satisfies $|v_j^{t+1}| \leq c_1||B_j||/||B||_F$, for some constant $c_1$.

**Bounding $dist(v^{t+1}, v^*)$:**

In Lemma C.4, set $\epsilon = \frac{||A^T B||}{2||A||||B||}\eta$ and $\delta = \frac{\eta}{2\tilde{r}}$, where $0 < \eta < 1$, then we have $\delta\epsilon||A||_F||B||_F \leq \frac{||A||_F||B||_F}{||A||||B||} \cdot \frac{\eta^2}{2\tilde{r}}||A^T B|| \leq \eta||A^T B||/2$, and $\epsilon||A||||B|| \leq \eta||A^T B||/2$. Therefore, with probability at least $1 - \gamma/T$, the following holds:

$$||(u^t)^T R_\Omega(\widetilde{M} - M_1) - (u^t)^T(M - M_1)|| \leq \eta||M - M_1||_F/\tilde{r} + \eta\sigma_1^*. \tag{15}$$

Hence, we have $||(u^t)^T R_\Omega(\widetilde{M} - M_1)|| \leq dist(u^t, u^*)||M - M_1|| + \eta||M - M_1||_F/\tilde{r} + \eta\sigma_1^*$.

Using the explicit formula for WAltMin update (see Eq.(46) and Eq.(47) in [3]), we can bound $\langle \hat{v}^{t+1}, v^* \rangle$ and $\langle \hat{v}^{t+1}, v^*_\perp \rangle$ as follows.

$$||\hat{u}^t|| \langle \hat{v}^{t+1}, v^* \rangle / \sigma_1^* \geq \langle u^t, u^* \rangle - \frac{\delta_1}{1 - \delta_1} \sqrt{1 - \langle u^t, u^* \rangle^2} - \frac{1}{1 - \delta_1} (\eta \frac{||M - M_1||_F}{\tilde{r} \sigma_1^*} + \eta).$$

$$||\hat{u}^t|| \langle \hat{v}^{t+1}, v^*_\perp \rangle / \sigma_1^* \leq \frac{\delta_1}{1 - \delta_1} \sqrt{1 - \langle u^t, u^* \rangle^2} + \frac{1}{1 - \delta_1} (dist(u^t, u^*) \frac{||M - M_1||}{\sigma_1^*} + \eta \frac{||M - M_1||_F}{\tilde{r} \sigma_1^*} + \eta).$$

As discussed in the end of Appendix C.2, we only need to consider the case when $||A^T B - (A^T B)_r||_F \leq \frac{1}{576\rho r^{1.5}} ||(A^T B)_r||_F$, where $\rho = \sigma_1^*/\sigma_r^*$. In the rank-1 case, this condition reduces to $||M - M_1||_F \leq \frac{\sigma^*}{576}$. For sufficiently small constants $\delta_1$ and $\eta$ (e.g., $\delta_1 \leq \frac{1}{20}$, $\eta \leq \frac{1}{20}$), and use the fact that $\langle u^t, u^* \rangle \geq \langle u^0, u^* \rangle$ and $dist(u^0, u^*) \leq 1/2$, we can further bound $\langle \hat{v}^{t+1}, v^* \rangle$ and $\langle \hat{v}^{t+1}, v^*_\perp \rangle$ as

$$||\hat{u}^t|| \langle \hat{v}^{t+1}, v^* \rangle / \sigma_1^* \geq \langle u^0, u^* \rangle - \frac{1}{10} \sqrt{1 - \langle u^0, u^* \rangle^2} - \frac{1}{10} \geq \frac{\sqrt{3}}{2} - \frac{2}{10} \geq \frac{1}{2}. \tag{16}$$

$$||\hat{u}^t|| \langle \hat{v}^{t+1}, v^*_\perp \rangle / \sigma_1^* \leq \frac{\delta_1}{1 - \delta_1} dist(u^t, u^*) + \frac{1}{576(1 - \delta_1)} dist(u^t, u^*) + \frac{1}{1 - \delta_1} (\eta \frac{||M - M_1||_F}{\tilde{r} \sigma_1^*} + \eta)$$

$$\overset{\xi_1}{\leq} \frac{1}{4} dist(u^t, u^*) + 2(\eta ||M - M_1||_F / \sigma_1^* + \eta), \tag{17}$$

where $\xi_1$ uses the fact that $\tilde{r} \geq 1$ and the assumption that $\delta_1$ is sufficiently small.

Now we are ready to bound $dist(v^{t+1}, v^*)$ as

$$dist(v^{t+1}, v^*) = \sqrt{1 - \langle v^{t+1}, v^* \rangle^2} = \frac{\langle \hat{v}^{t+1}, v^*_\perp \rangle}{\sqrt{\langle \hat{v}^{t+1}, v^*_\perp \rangle^2 + \langle \hat{v}^{t+1}, v^* \rangle^2}} \leq \frac{\langle \hat{v}^{t+1}, v^*_\perp \rangle}{\langle \hat{v}^{t+1}, v^* \rangle}$$

$$\overset{\xi_1}{\leq} \frac{1}{2} dist(u^t, u^*) + 4(\eta ||M - M_r||_F / \sigma_1^* + \eta), \tag{18}$$

where $\xi_1$ follows from substituting Eqs. (16) and (17). Rescaling $\eta$ as $\eta/4$ gives the desired bound of Lemma C.2 for the rank-1 case. Rank-r proof can be obtained by following a similar framework.

**Bounding $v_j^{t+1}$:**

In this step, we need to prove that the $j$-th entry of $v^{t+1}$ satisfies $|v_j^{t+1}| \leq c_1 \frac{||B_j||}{||B||_F}$ for all $j$, under the assumption that $u^t$ satisfies the norm bound $|u_i^t| \leq c_1 \frac{||A_i||}{||A||_F}$ for all $i$.

The proof follows very closely to the second part of proving Lemma C.3 in [3], except that an extra multiplicative term $(1 + \epsilon)$ will show up when bounding $\sum_i \delta_{ij} w_{ij} u_i^t \widetilde{M}_{ij}$ using Bernstein inequality. More specifically, let $X_i = (\delta_{ij} - \hat{q}_{ij}) w_{ij} u_i^t \widetilde{M}_{ij}$. Note that if $\hat{q}_{ij} = 1$, then $\delta_{ij} = 1$, $X_i = 0$, so we only need to consider the case when $\hat{q}_{ij} < 1$, i.e., $\hat{q}_{ij} = q_{ij}$, where $q_{ij}$ is defined in Eq.(1).

Suppose $\Pi$ is fixed and its dimension satisfies $k = \Omega(\frac{\log(n)}{\epsilon^2})$, then according to Lemma B.6, we have that w.h.p. in $n$,

$$|\widetilde{M}_{ij}| \leq |M_{ij}| + \epsilon ||A_i|| \cdot ||B_j|| \leq (1 + \epsilon) ||A_i|| \cdot ||B_j||, \quad \forall (i, j). \tag{19}$$

Hence, we have

$$\frac{\widetilde{M}_{ij}^2}{\hat{q}_{ij}} \overset{\xi_1}{\leq} \frac{(1 + \epsilon)^2 ||A_i||^2 ||B_j||^2}{m \cdot (\frac{||A_i||^2}{2n||A||_F^2} + \frac{||B_j||^2}{2n||B||_F^2})} \leq \frac{2n(1 + \epsilon)^2}{m} \cdot ||B_j||^2 ||A||_F^2, \tag{20}$$

$$\frac{(u_i^t)^2}{\hat{q}_{ij}} \overset{\xi_2}{\leq} \frac{c_1^2 ||A_i||^2 / ||A||_F^2}{m \cdot (\frac{||A_i||^2}{2n||A||_F^2} + \frac{||B_j||^2}{2n||B||_F^2})} \leq \frac{2nc_1^2}{m}, \tag{21}$$

where $\xi_1$ follows from substituting Eqs.(19) and (1), and $\xi_2$ follows from the assumption that $|u_i^t| \leq c_1 ||A_i|| / ||A||_F$.

We can now bound the first and second moments of $X_i$ as

$$|X_i| \leq |w_{ij} u_i^t \widetilde{M}_{ij}| \leq \sqrt{\frac{(u_i^t)^2}{\hat{q}_{ij}}} \sqrt{\frac{\widetilde{M}_{ij}^2}{\hat{q}_{ij}}} \overset{\xi_1}{\leq} \frac{2nc_1(1+\epsilon)}{m} ||B_j|| ||A||_F.$$

$$\sum_i Var(X_i) = \sum_i \hat{q}_{ij}(1-\hat{q}_{ij}) w_{ij}^2 (u_i^t)^2 \widetilde{M}_{ij}^2 \leq \sum_i \frac{(u_i^t)^2}{\hat{q}_{ij}} (1+\epsilon)^2 ||A_i||^2 ||B_j||^2$$

$$\overset{\xi_2}{\leq} \frac{2nc_1^2(1+\epsilon)^2}{m} ||B_j||^2 ||A||_F^2,$$

where $\xi_1$ and $\xi_2$ follows from substituting Eqs.(20) and (21).

The rest proof involves applying Bernstein's inequality to derive a high-probability bound on $\sum_i X_i$, which is almost the same as the second part of proving Lemma C.3 in [3], so we omit the details here. The only difference is that, because of the extra multiplicative term $(1+\epsilon)$ in the bound of the first and second moments, the lower bound on the sample complexity $m$ should also be multiplied by an extra $(1+\epsilon)^2$ term. By restricting $0 < \epsilon < 1/2$, this extra multiplicative term can be ignored as long as the original lower bound of $m$ contains a large enough constant. □

## C.4   Proof of Theorem 3.1

We now prove our main theorem for rank-1 case here. Rank-$r$ proof follows a similar line of reasoning and can be obtained by combining the current proof with the rank-$r$ analysis in the original proof of LELA [3]. Similar to the previous section, we use $\widehat{u}^t$ and $\widehat{v}^{t+1}$ to denote the $t$-th and $(t+1)$-th step iterates (which are column vectors in this case) of the WAltMin algorithm. Let $u^t$ and $v^{t+1}$ be the corresponding normalized vectors.

The closed form solution for WAltMin update at $t+1$ iteration is

$$||\widehat{u}^t|| \widehat{v}_j^{t+1} = \sigma_1^* v_j^* \frac{\sum_i \delta_{ij} w_{ij} u_i^t u_i^*}{\sum_i \delta_{ij} w_{ij} (u_i^t)^2} + \frac{\sum_i \delta_{ij} w_{ij} u_i^t (\widetilde{M} - M_1)_{ij}}{\sum_i \delta_{ij} w_{ij} (u_i^t)^2}.$$

Writing in matrix form, we get

$$||\widehat{u}^t|| \widehat{v}_j^{t+1} = \sigma_1^* \langle u^*, u^t \rangle v^* - \sigma_1^* B^{-1}(\langle u^*, u^t \rangle B - C) v^* + B^{-1} y, \tag{22}$$

where $B$ and $C$ are diagonal matrices with $B_{jj} = \sum_i \delta_{ij} w_{ij} (u_i^t)^2$ and $C_{jj} = \sum_i \delta_{ij} w_{ij} u_i^t u_i^*$, and $y$ is the vector $R_\Omega(\widetilde{M} - M_1)^T u^t$ with entries $y_j = \sum_i \delta_{ij} w_{ij} u_i^t (\widetilde{M} - M_1)_{ij}$.

Each term of Eq.(22) can be bounded as follows.

$$||(\langle u^*, u^t \rangle B - C) v^*|| \leq dist(u^t, u^*), \quad ||B^{-1}|| \leq 2, \tag{23}$$

$$||y|| = ||R_\Omega(\widetilde{M} - M_1)^T u^t|| \overset{\xi_1}{\leq} dist(u^t, u^*) ||M - M_1|| + \eta ||M - M_1||_F / \tilde{r} + \eta \sigma_1^*, \tag{24}$$

where $\xi_1$ follows directly from Lemma C.4. The proof of Eq.(23) is exactly the same as the proof of Lemma B.3 and B.4 in [3].

According to Lemma C.2, since the distance is decreasing geometrically, after $O(\log(\frac{1}{\zeta}))$ iterations we get

$$dist(u^t, u^*) \leq \zeta + 2\eta ||M - M_1||_F / \sigma_1^* + 2\eta. \tag{25}$$

Now we are ready to prove the spectral norm bound in Theorem 3.1:

$$||M_1 - \widehat{u}^t(\widehat{v}^{t+1})^T||$$
$$\leq ||M_1 - u^t(u^t)^T M_1|| + ||u^t(u^t)^T M_1 - \widehat{u}^t(\widehat{v}^{t+1})^T||$$
$$\leq ||(I - u^t(u^t)^T)M_1|| + ||u^t[(u^t)^T M_1 - ||\widehat{u}^t||(\widehat{v}^{t+1})^T]||$$
$$\overset{\xi_1}{\leq} \sigma_1^* dist(u^t, u^*) + ||\sigma_1 \langle u^t, u^* \rangle v^* - ||\widehat{u}^t||(\widehat{v}^{t+1})^T||$$
$$\overset{\xi_2}{\leq} \sigma_1^* dist(u^t, u^*) + ||\sigma_1^* B^{-1}(\langle u^*, u^t \rangle B - C)v^*|| + ||B^{-1}y||$$
$$\overset{\xi_3}{\leq} \sigma_1^* dist(u^t, u^*) + 2\sigma_1^* dist(u^t, u^*) + 2 dist(u^t, u^*)||M - M_1|| + 2\eta||M - M_1||_F/\tilde{r} + 2\eta\sigma_1^*$$
$$\overset{\xi_4}{\leq} 5(\zeta\sigma_1^* + 2\eta||M - M_1||_F + 2\eta\sigma_1^*) + 2\eta||M - M_1||_F + 2\eta\sigma_1^*$$
$$= 5\zeta\sigma_1^* + 12\eta||M - M_1||_F + 12\eta\sigma_1^* \tag{26}$$

where $\xi_1$ follows from the definition of $dist(u^t, u^*)$, the fact that $||u^t|| = 1$, and $(u^t)^T M_1 = \sigma_1 \langle u^t, u^* \rangle v^*$, $\xi_2$ follows from substituting Eq.(22), $\xi_3$ follows from Eqs.(23) and (24), and $\xi_4$ follows from the Eq.(25), and fact that $||M - M_1|| \leq \sigma_1^*$, $\tilde{r} \geq 1$. Rescaling $\zeta$ to $\zeta/(5\sigma_1^*)$ (this will influence the number of iterations) and also rescaling $\eta$ to $\eta/12$ gives us the desired spectral norm error bound in Eq.(7). This completes our proof of the rank-1 case. Rank-$r$ proof follows a similar line of reasoning and can be obtained by combining the current proof with the rank-$r$ analysis in the original proof of LELA [3].

## C.5 Sampling

We describe a way to sample $m$ elements in $O(m \log(n))$ time using distribution $q_{ij}$ defined in Eq. (1). Naively one can compute all the $n^2$ entries of $\min\{q_{ij}, 1\}$ and toss a coin for each entry, which takes $O(n^2)$ time. Instead of this binomial sampling we can switch to row wise multinomial sampling. For this, first estimate the expected number of samples per row $m_i = m(\frac{||A_i||^2}{2||A||_F^2} + \frac{1}{2n})$. Now sample $m_1$ entries from row 1 according to the multinomial distribution,

$$\widetilde{q}_{1j} = \frac{m}{m_1} \cdot (\frac{||A_1||^2}{2n||A||_F^2} + \frac{||B_j||^2}{2n||B||_F^2}) = \frac{\frac{||A_1||^2}{2n||A||_F^2} + \frac{||B_j||^2}{2n||B||_F^2}}{\frac{||A_i||^2}{2||A||_F^2} + \frac{1}{2n}}.$$

Note that $\sum_j \widetilde{q}_{1j} = 1$. To sample from this distribution, we can generate a random number in the interval $[0, 1]$, and then locate the corresponding column index by binary searching over the cumulative distribution function (CDF) of $\widetilde{q}_{1j}$. This takes $O(n)$ time for setting up the distribution and $O(m_1 \log(n))$ time to sample. For subsequent row $i$, we only need $O(m_i \log(n))$ time to sample $m_i$ entries. This is because for binary search to work, only $O(m_i \log(n))$ entries of the CDF vector needs to be computed and checked. Note that the specific form of $\widetilde{q}_{ij}$ ensures that its CDF entries can be updated in an efficient way (since we only need to update the linear shift and scale). Hence, sampling $m$ elements takes a total $O(m \log(n))$ time. Furthermore, the error in this model is bounded up to a factor of 2 of the error achieved by the Binomial model [7] [21]. For more details please see our Spark implementation.

## D Related work

**Approximate matrix multiplication:** In the seminal work of [14], Drineas et al. give a randomized algorithm which samples few rows of $A$ and $B$ and computes the approximate product. The distribution depends on the row norms of the matrices and the algorithm achieves an additive error proportional to $||A||_F ||B||_F$. Later Sarlos [29] propose a sketching based algorithm, which computes sketched matrices and then outputs their product. The analysis for this algorithm is then improved by [10]. All of these results compare the error $||A^T B - \tilde{A}^T \tilde{B}||_F$ in Frobenius norm.

For spectral norm bound of the form $||A^T B - C||_2 \leq \epsilon ||A||_2 ||B||_2$, the authors in [29, 11] show that the sketch size needs to satisfy $O(r/\epsilon^2)$, where $r = rank(A) + rank(B)$. This dependence on rank is later improved to stable rank in [26], but at the cost of a weaker dependence on $\epsilon$. Recently,

Cohen et al. [12] further improve the dependence on $\epsilon$ and give a bound of $O(\tilde{r}/\epsilon^2)$, where $\tilde{r}$ is the maximum stable rank. Note that the sketching based algorithm does not output a low rank matrix. As shown in Figure 2, rescaling by the actual column norms provide a better estimator than just using the sketched matrices. Furthermore, we show that taking SVD on the sketched matrices gives higher error rate than our algorithm (see Figure 3(b)).

**Low rank approximation:** [16] introduced the problem of computing low rank approximation of a given matrix using only few passes over the data. They gave an algorithm that samples few rows and columns of the matrix and computes its SVD for low rank approximation. They show that this algorithm achieves additive error guarantees in Frobenius norm. [15, 29, 19, 13] have later developed algorithms using various sketching techniques like Gaussian projection, random Hadamard transform and volume sampling that achieve relative error in Frobenius norm.[35, 28, 18, 6] improved the analysis of these algorithms and provided error guarantees in spectral norm. More recently [11] presented an algorithm based on subspace embedding that computes the sketches in the input sparsity time.

Another class of methods use entrywise sampling instead of sketching to compute low rank approximation. [1] considered an uniform entrywise sampling algorithm followed by SVD to compute low rank approximation. This gives an additive approximation error. More recently [3] considered biased entrywise sampling using leverage scores, followed by matrix completion to compute low rank approximation. While this algorithm achieves relative error approximation, it takes two passes over the data.

There is also lot of interesting work on computing PCA over streaming data under some statistical assumptions, e.g., [2, 27, 5, 30]. In contrast, our model does not put any assumptions on the input matrix. Besides, our goal here is to get a low rank matrix and not just the subspace.