[Reviews · NeurIPS 2016]

Reviewer 1

Summary

Given two matrices A and B, the authors study the problem of constructing a low rank approximation to the matrix C = A'B, in the streaming setting. The results are somewhat interesting but I am not convinced this is an interesting problem.

Qualitative Assessment

motive the problem better

Confidence in this Review

2-Confident (read it all; understood it all reasonably well)


Reviewer 2

Summary

The main contribution in this paper, as I see it, it to refine the random projection based matrix product approximation technique. Intuitively, when multiplying two randomly projected matrices there are two sources of error, the distortion of column norms and distortion of the angle between them. The authors astutely note that the first error type is avoidable without much computational overhead. The experimental result corroborate that this is indeed beneficial.

Qualitative Assessment

This paper presents a technique that should receive more mathematical attention. It is a simple yet effective modification of a standard method. The main drawback of this paper is the algorithm analysis which is rather weak. Nevertheless, it provides a baseline upon future work could improve.

Confidence in this Review

2-Confident (read it all; understood it all reasonably well)


Reviewer 3

Summary

This paper gives an improved method for obtaining a low rank approximation of a matrix product. Instead of directly applying low-rank sketches to both matrices and then multiplying them, the method picks low rank objects from the product from the norms of the columns of the sketches, and uses the resulting sum as an approximation instead. The paper demonstrates the utility of this method by both giving theoretical proofs for its error guarantees, and giving a distributed implementation of it in Spark that demonstrates a factor 2~3 performance gain over the LELA method.

Qualitative Assessment

The theoretical question addressed is interesting, and the analysis that demonstrates gains over direct sketch-and-multiply is rather intricate. The experiments are conducted on large data sets and in distributed settings. They demonstrate the feasibility of the proposed methods, but at the same time also show a fairly large error of all the methods proposed. The choice of data sets is probably the biggest weakness of this paper: naive matrix multiplication takes dn^2 time, so on all 3 of the data sets studied, a brute force method is only 4~5x slower. Furthermore, the low dimensional nature of the URL benchmarks (compared to sketch size) raises the possibility that the gains are coming from dimensionality reductions in the matrices beforehand. It would also be useful to compare this method with sketch-and-multiply, or computing low rank approximations of both terms and multiplying them, both theoretically and experimentally.

Confidence in this Review

2-Confident (read it all; understood it all reasonably well)


Reviewer 4

Summary

The authors have proposed a novel single pass PCA algorithm for computing the low-rank approximation of the inner product of two matrices A and B. Therefore, two smaller representations of the full-state matrices are computed using matrix sketching. Then, based on the sketched matrices, an estimator for the inner product is computed. The key innovation of the algorithm is the rescaled Johnson–Lindenstrauss (JL) embedding, used to estimate the important entries of the inner product. Finally, a matrix completion algorithm is used to obtain the desired rank-r approximation. In addition to theoretical results, the authors provide an Apache Spark implementation and empirical evaluations using both synthetic and real data.

Qualitative Assessment

The paper is well written, and the theoretically developments seem sound and interesting, though I checked the proofs in the appendix only briefly. The provided code is structured and well readable. Technical quality: Overall the technical quality is very good. My main concern is with the experiments, specifically the algorithms run-time, shown in Figure (3a). The evaluation seems somewhat fuzzy for two reasons. (1) The sketch size 'k=10' is used, while in Table (1) the sketch size 'k=4000,8000' are shown. (2) The authors excluding the time for executing alternating least squares (ALS), because they claim that this is a common computational step. However, ALS is an iterative algorithms and the time to converge depends considerably on the initial input. Hence, I could assume that the proposed algorithms requires several more iterations to converge. Thus, the algorithm might not be that much faster in practice. I hope this issue can be clarified by the authors. Moreover, a Figure which shows the run-time vs. sketch size and in addition error vs. sketch size for the same data-set would be insightful. Originality: The theoretical results seem to be different from prior work. In particular the work on rescaled JL embedding is interesting. Potential impact or usefulness: The usefulness of the proposed algorithm relies mainly on the compute time. However as stated above, the computational gain (in practice) seems somewhat fuzzy. A better illustration of the trade-off between time and error would be useful. Clarity and presentation: Overall the presentation is very good. The paper is well written. However some of the Figures are non vector graphics, e.g., 1 and 3a. Hence, the Figures look a bit blurry on the printed document or when zooming in.

Confidence in this Review

2-Confident (read it all; understood it all reasonably well)


Reviewer 5

Summary

The paper considers approximating PCA of A’*B, where A and B are large matrices. The advantages of the proposed algorithm is that 1) it only loads the data matrices once into the memory from disk (single pass) and 2) it provides guarantees of the approximation accuracy. I think the method is interesting and the paper is very well-written. The SPARK implementation is also appreciated.

Qualitative Assessment

, I have several concerns: The major difference between the algorithm here and the algorithm in [1] (i.e., LELA) is a sketching step. The sketching step is prone to lose information and degrades the performance (as seen in Fig.3 (b)). It also requires considerable computational resource if A and B are not sparse. Therefore, it is not very clearly seen that the proposed approach is preferable over LELA. I would recommend the authors to find some examples where sketching brings more significant time improvement. For the datasets under test, n=1000 ~ 30000 are not large. A natural question is how large n can be handled by the proposed algorithm. As a motivating example, the authors talked about CCA and A’* B being cross-correlation matrix. There are many cases in CCA applications that both dimensions of A (B) are very large (with similar dimensions). Under such cases, is it still efficient to use the proposed algorithm? Which step will be the bottleneck? The authors are suggested to comment on these aspects.

Confidence in this Review

2-Confident (read it all; understood it all reasonably well)


Reviewer 6

Summary

The paper proposed a new algorithm called SMP-PCA for computing a low-rank approximation of the product of A^T times B. It is time efficient by taking only a single pass of the two matrices A and B. The main novelty of the algorithm lies in utilizing a rescaled JL embedding to produce more accurate approximation. The experimental result showed that SMP-PCA achieved approximately half the running time compared to LELA, but with larger errors. It indeed shows that SMP-PCA outperformed the method of doing SVD on the sketched matrices.

Qualitative Assessment

This paper developed an new algorithm that achieves time efficiency. But the novelty in the paper is limited. In the three-step algorithm, the paper developed a rescaled JL embedding scheme to achieve better approximation (step 2). The other two steps, i.e., compute sketches and implementing matrix completion, however, are pretty standard techniques. The reviewer also expected to see more solid analysis behind the algorithm: 1. In the discussion of the advantage of rescaled JL embedding over the naive JL embedding, the paper only explained the intuition behind it but lacked systematic proof, which made it more like a heuristic. 2. The reviewer is wondering if WAltMin algorithm in step 3 can be replaced by other matrix completions algorithms. The paper did not explain the logic of using WAltMin or provide analysis or experimental results of using other matrix completion algorithms as a comparison. The experimental results showed that SMP-PCA achieved approximately half the running time compared to LELA but suffered from larger error. The reviewer was not fully convinced of the practical applications of SMP-PCA unless the authors gave practical examples where speed is more emphasized than the accuracy. In the comparison of SMP-PCA and the method of doing SVD on the sketched matrics, the reviewer did not see the logic of generating column vectors of A and B from a with angel theta. In real data applications, what does it imply to have theta go to zero?

Confidence in this Review

2-Confident (read it all; understood it all reasonably well)